# Mixed Assessment of Virtual Serious Games Applied in Architectural and Urban Design Education

**DOI:** 10.3390/s21093102

**Published:** 2021-04-29

**Authors:** David Fonseca, Janaina Cavalcanti, Enric Peña, Victor Valls, Mónica Sanchez-Sepúlveda, Fernando Moreira, Isidro Navarro, Ernesto Redondo

**Affiliations:** 1Architecture La Salle, Universitat Ramon Llull, 08022 Barcelona, Spain; enric.pc@salle.url.edu (E.P.); victor.valls@salle.url.edu (V.V.); monica.sanchez@salle.url.edu (M.S.-S.); 2Instituto de Investigación e Innovación en Bioingeniería (I3B), Universitat Politècnica de Valencia, 46022 Valencia, Spain; cjanaina@gmail.com; 3REMIT, IJP, Universidade Portucalense, 4200-072 Porto, Portugal; fmoreira@uportu.pt; 4IEETA, Universidade de Aveiro, 3810-193 Aveiro, Portugal; 5Escola Tècnica Superior d’Arquitectura de Barcelona, Universitat Politècnica de Catalunya, 08028 Barcelona, Spain; isidro.navarro@upc.edu (I.N.); ernesto.redondo@upc.edu (E.R.)

**Keywords:** virtual reality, interactive systems, serious games, mixed assessment, user experience, user profile, usability, learning implementation, architecture and urban projects, decision making

## Abstract

The creation and usage of serious games on virtual reality (VR) and/or interactive platforms for the teaching of architecture, construction, urban planning, and other derived areas, such as security and risk prevention, require design processes, studies, and research that lead to further consolidation expansion. In that sense, this paper presents two main aims developed: the improvement of a virtual navigation system through the results of previous user studies and mixed research (quantitative and qualitative) improved based on the user perception for educational and professional uses. The VR system used is based on Unreal Engine programming of the HTC Vive sensor. This study is related to the GAME4City 3.0 and a broader project focused on gamified visualization and its educational uses in architectural and urban projects. The results reflect great interest, good usability, and high motivation for further usage for all types of users. However, an apparent resistance to deepen its use continues to be perceived in academia. Based on the research results, weak points of educational gamified systems have been identified, and the main differences and needs in user profiles’ function. With these data, progress regarding implementing this kind of system at the teaching and professional levels must be pursued.

## 1. Introduction

We are in a time of changes present at all levels, i.e., social, relational, technological, educational, etc., many of which have been reinforced by the world crisis related to the COVID-19 pandemic. Before the confinement (March 2020), the presence of many activities was obligatorily adapted and forced to migrate to adjust to the new situation. Formal aspects of face-to-face interactions linked to physical content have been replaced overnight with digital content. This replacement has generated the necessity to adapt these physical formats to new user profiles and technologies [1,2,3,4].

Education was one of the sectors that have been changing increasingly abruptly [5]. The forced migration from face-to-face to online contexts has transformed how teachers transfer their knowledge to students and deal with content and subjects for their competency validation. Digital systems, especially interactive systems, together with online work, allow a much faster, cheaper and more sustainable workflow than traditional methods. The printed plans, panels, models, and physical deliveries of architecture, urban design, civil, and building engineering studies have been in crisis due to students’ confinement and dislocation due to the COVID-19 pandemic. The identified advantages of virtual strategies are clear and have been extensively documented [6,7,8,9]. Nevertheless, their degree of implementation at the educational level is perceived as low, even when the technologies or methodologies are of interest to the end-user or necessary for professional use.

This paper’s novelty lies in identifying and improving (based on the results of previous phases of the project) the critical aspects that affect the usability and space comprehension in the user’s interaction with VR environments applied to architectural and urban design. Virtually exploring scenes that in reality would not be possible (for example, non-constructed building and/or urban environments) stimulates the student to understand better and memorize the academic content (in the architecture framework, the relation of the proposal with the scale of the environment, the materials, textures, illumination, etc.) [10,11,12]. However, from programming with Unreal Engine for a visualization system based on the HTC Vive sensor, those negative aspects that diminished comfort and satisfaction in its use have been improved: the feeling of sickness, the quality of textures and materials of the virtual elements (critical for architecture representation), the selection of interactive elements through a new menu, the improvement of the handheld sensor information for decision making, and the relative position of the virtual avatar without changing the real-physical position of the user between the optical sensors of the device. Other actions have also been redesigned and optimized, such as selecting and placing elements on the space, access to the object library, and the manipulation of urban elements through user interactions [13].

In this sense, the HTC Vive device consists of a headset with two controllers and two infrared laser emitter units for tracking called Lighthouses. The two hand-sensors have different buttons that can be programmed. Visual aids have been integrated so that when they are displayed within the VR environment, they have information about their actions. On the other hand, two 1080 × 1200 pixel displays updated at 90 Hz cover a nominal field of view of around 110°, and the laser units are two boxes that alternately send out horizontal and vertical infrared laser sweeps spanning 120° in each direction. The difference in time at which the laser hits the various photodiodes allows the recovery of the position and orientation of the headset [14]. The HTC Vive is designed to track an observer who freely moves through a space of up to 4 × 4 m^2^, a larger space than offered by the other widely used VR system, Oculus Rift.

This approach is novel insofar as it is applied to enhance the training of students in architecture, an educational field with very few ongoing experiences in the use of VR and gamified approaches to support continuous learning [15,16,17,18,19,20]. This area has historically been poorly evaluated, unlike in other educational fields such as medicine, science, and even engineering, where user studies with students and/or educational innovations are frequently assessed. In this framework, the few existing types of research usually use quantitative approaches on specific case studies. However, given the growing need for digitization and interaction with complex models, our work identifies critical aspects of programming and the user’s perception of the proposed models combining quantitative and qualitative data, following our innovative previous research [21,22].

The well-known mixed assessment methods bring a degree of novelty to our study when applied to architecture and urban studies, where the academic usability and assessment studies are not usually performed [23]. The mixed approach has allowed us to obtain greater precision in analyzing the data collected, inviting a deep reflection on the users concerning the technology evaluated and its use [24]. Identifying the different perceptions according to the user profile will allow us to personalize virtual environments better, optimize the user experience, and, ultimately, the academic processes. In conclusion, we are faced with an approach to educational analytics processes from mixed research in an academic environment undergoing deep and ongoing organizational change.

Starting from the idea that the Game4City 3.0 project seeks to implement new gamified virtual models for learning and understanding architectural and urban projects, this paper aims to solve the following Research Questions:RQ1: What are the main problems in VR sensor programming using the main videogame engines such as Unity or Unreal?RQ2: Are there differences in use, perception, satisfaction and/or expectations of serious games implemented in VR systems in the function of the user’s profile?RQ3: Why are the immersive and/or gamified systems poorly used in architectural education when their potential has been widely demonstrated?

Using a Project-Based Learning (PBL) approach, the Game4City 3.0 project and its sub-projects (ArchGame4City and EduGame4City) are focused on designing new smart and collaborative approaches for tactical cities, building design, and urban projects. As we have previously commented, the novelty is based on the use of technological ecosystems, interactive interfaces, and gamified methodologies to improve the learning and comprehension of complex spaces by the students and end-users. Based on the results of the systematic user experience approach, we will establish relationships with the motivation, utility and usability perceived by all agents (teachers, students, professionals, and citizens) in the eco-design of new urban/construction/training tactical projects, thereby completing previous case studies conducted within the same context.

To resolve the Research Questions defined, we have reviewed in Section 2 some issues related to the main problems and challenges of using VR sensors for educational purposes and how it is possible to increase their usability and the level of satisfaction of the end-users. In the same section, we identify new smart city designs using new approaches for projects and other issues related to architectural concepts (for example, risk management). The use of gamification, i.e., the use of interactive systems by end-users (students or citizens), is shown to influence the projects’ development and usability. Finally, we have introduced the Game4City 3.0 project, where this research is enclosed. In Section 3, we have addressed RQ1. Based on the summary of the previous studies related to the project (to identify the main issues to correct in this phase), we present our methodology to improve the VR system’s usability and how to assess it. In Section 4, we present the mixed assessment results for addressing RQ2 and RQ3 to clarify the differences between users’ motivation for addressing and implementing virtual gamified systems in a more comprehensive view. A final section consisting of discussions and conclusions closes this paper.

## 2. Theoretical Framework

### 2.1. Virtual Reality Systems: Main Issues

The usage of VR systems for both student and professional training is a process that has been widely documented in all kinds of areas [25,26,27]. The wide variety of available sensors and the increasing ease of the design and programming of new devices have allowed the implementation of different solutions. In the vast majority of cases, the programming of the sensor has been carried out, considering individual project specifications due to the lack of standards [28,29,30]. The particular solutions developed have been determined to improve specific practices and/or subjects at the teaching or professional level [31,32,33], including architecture [34,35,36,37]. In all these practices, one key concept of VR systems is the “presence”, which has been defined as “the sense of ‘being’ in the VR world” [38]. We further describe that presence is related to how much the user “feels” that the VR is real (a more substantial feeling indicates a stronger sense of presence, while the opposite weak feeling indicates a more fragile presence). Presence has the two following characteristics that are essential for positive usability:Immersion (a psychological state characterized by perceiving oneself to be enveloped by, included in, and interacting with an environment that provides a continuous stream of stimuli and experiences) [39];Involvement (an individual’s psychological state, energy focus, and attention on their own set of stimuli, [40]).

If users focus more attention on the VR environment, consequently, they become more involved. However, if a user is preoccupied with personal problems, he/she will become less involved. One level of involvement depends on the degree of significance or meaning that the individual attaches to the stimuli, activities, or events. An essential concept of presence is based on the continuities, connectedness, and coherence of the stimulus flow [39]. Many crafters have been analyzed to program VR environments and devices more realistically. Such analysis includes variable contents, technological and emotional artifices and design strategies that combine action, symbolic, and sensory factors [41]. Studies have suggested that stress, fear, and anxiety lead to a stronger sense of presence [42]. However, at the same time, anxiety could diminish a user’s performance on complex tasks [43]. The following needs are identified to consider the possible “usage models” [44]:Free and interactive navigation and exploration of the virtual 3D environment with access to teaching multimedia content;Instructional presentations to demonstrate how to perform a specific task or data;Guided simulations with alerts, which notify users of the occurrence of mistakes and guide them to correct their results;Free performance with the result at the end of the simulation.

Some authors have suggested that the last point is the most efficient mode for training, based on constructivism theory, which values the learner’s active presence when constructing the learning process [45,46]. Other researchers have affirmed that to describe fixed stories (such as procedures), non-interactive storytelling is better [47]. It is essential to elucidate that the latter authors consider non-interactive variables that could standardize interactions (such as the selection of objects and dynamic menus). Work was developed to evaluate the influences of degrees of freedom in different tasks in a virtual environment related to risk and security, considering the scope of risk and danger in 2013. The authors [48] realized that whether the degree of freedom exerted influence depended on the task at hand. Besides, it is no less essential to consider user characteristics [49], such as age, cultural variables, gender, etc. This is because of the ability to perceive risk changes or attitudes in VR environments by individuals due to their beliefs, motivations, and relationships with other people.

On the other hand, we can identify three major problems that affect these systems and seem fundamental in the current lack of implementation at the teaching/learning level:Motion sickness (inherent to programming in a virtual environment);Price;Player interaction and engagement with the system.

In VR, users can experience motion sickness symptoms, referred to as VR sickness or cyber-sickness. The related symptoms include, but are not limited to, eye fatigue, disorientation, and nausea, which can impair users’ VR experience [50]. A variety of factors can cause these symptoms as follows:Sensor/device type. There are two main types of sensors and displays for VR headsets: CAVE systems (project the scenery on the walls of a room with a motion tracker that monitors the user’s position and changes the projected images accordingly) and HMD (head-mounted displays) systems (these are the standard headset devices that, instead of projecting the scenery onto the walls, have screens in front of the user’s eyes). Due to the differences between the perceived space and reality and the higher degree of isolation, HMD systems tend to be more disorienting than CAVE systems [51].Field of view (FOV). Adjusting the maximum visual angle of display that the system provides, primarily by reducing it, alleviates the user’s discomfort, particularly during camera movement. With a lower FOV, the view becomes narrower and resembles that of a telescope, which makes the view more sensitive to headset movements and requires the user to concentrate on not moving his/her head to watch the contents properly [52].Latency. This variable is related to the window of time between the user’s movement in real life and a pixel’s time resulting from that movement response [53,54]. Accordingly, the lower the latency, the lower the lack of time, and the less discomfort the user experiences [55].Flicker. This occurs when the headset screen shines unsteadily and/or varies rapidly in brightness, leading to severe VR sickness for the user [56]. The level of flicker can be influenced by the display’s refresh rate, luminance, and FOV.Optical flow. A significant hurdle of virtual reality is completely disconnecting what the user sees with what the user is feeling. If the user is afflicted with an illusory movement, this can cause him/her to experience a strong sense of “vection” and VR sickness. Ironically, users have reported having the highest level of discomfort while in a ten m/s motion but maintaining (or slightly decreasing) their motion sickness while moving at up to 60 m/s due to their feeling of presence being weakened.Graphic realism. While sharper graphics help with the flicker level and reduce discomfort, higher visual fidelity with VR content does not necessarily lower the user’s discomfort level. In contrast, due to the view being more realistic, the user may experience a higher level of sensory discrepancy between visual and vestibular information, leading to higher levels of VR sickness [57].Controllability. We can divide the controllability between active and passive experiences, depending on how they engage with VR media. Passive navigation does not simulate a person’s movement and/or does not give full control to the player, which forces the user to engage passively with the media, thereby severely aggravating his/her VR sickness.

The major hurdle that impedes VR systems from becoming mainstream is their price. Despite the quality significantly improving and the prices frequently lowering (which is reflected by the growth of the sales [58,59]), the price for a VR headset is comparable to a new-generation console or a mid-tier computer [60,61]. However, why do we not treat VR headsets like any other console? After all, their prices are pretty comparable. The answer is that, unlike a console, a VR headset usually needs a supplementary system to work, which adds to its cost [62]. To use a PlayStation VR headset, you need a PlayStation 4 or superior system [63]. Besides, to use an Oculus Rift [64], an HTC VIVE [65] or a VALVE Index [66], you need a high-end PC, which significantly increases the price one has to pay to use this technology.

Another critical factor to consider is all the hurdles related to the user starting to use VR, which we can relate to player interaction and engagement. Unlike a regular computer or console game, VR cannot be accessed at small windows of time (or at the very least, it is not comfortable to do so) for a variety of reasons, including the following:Space: The point of VR is for the user to be able to feel like he/she is inside the platform; thus, it is necessary to make some space for the user to turn around, move the hand controllers, and even move around slightly. This leads to the need to have an open area available for moving around, i.e., a space of 2 M × 1.5 M as a minimum for a comfortable experience [67].Sockets: While it is true that VR hand controllers have internal batteries, the user still must deal with the cables running from the headset to the computer (with some exceptions), which restrict the user’s freedom of movement and the connection of the sensors to both the AC sockets and the computer.Disconnect: The VR headset, unlike any other current mainstream console, works by isolating the user’s sensory attention. The user stops being able to engage with anything other than the VR media because he/she cannot see or hear anything outside of the system. Simultaneously, efforts are being made to control the remaining senses to completely isolate the user to give him/her a more realistic sensation [68,69].

### 2.2. Learning Processes, ICTs for Quick-Response Systems and New Active Methodologies

In the guidelines defined by The European Higher Education Area (EHEA), the most critical aspect of the teaching–learning process is updating the competencies that the students must acquire in their studies [70]. Based on the virtualization caused by the global COVID-19 pandemic and the subsequent migration to hybrid systems, new approaches which are more dynamic, sustainable, digital and virtual have been implemented for both professional and educational needs [71,72,73]. These processes have changed the promotion of knowledge, communication behaviors, customs, and ways of thinking [74].

The capacity of spatial vision and a graphic representation can be identified as the most critical skills that can be acquired by students of architecture, building design, civil engineering or urban design. Promoting these skills is a commonly used project-based learning (PBL) approach and combining classic methods (2D printed layouts and physical mock-ups) with computer-aided design (CAD) applications [75]. In this sense, there has been a significant shift in recent decades, particularly as a result of the industry’s professional needs:It is increasingly common for any professional project to be endorsed/certified using BIM (building information modeling) systems [56];The quality of the objects, textures, lighting, and materials that can be viewed in actual three-dimensional models has reached a level of reality that makes it possible to make far more precise and competent decisions than those obtained through plans and models;To understand three-dimensional space, the end-user is increasingly demanding and requests more real and explicit methods.

These strategies are aligned with the industry’s most popular methodologies, such as PBL and teamwork [58]. In particular, BIM systems are adapted to the project’s collaborative monitoring [76,77,78,79] and integrate applications [80,81], either internally or externally, to speed up the application of hyper-realistic elements [82,83,84]. In this sense, through real-time rendering engines, we can visualize and assimilate the design of the space in a faster and more concise way than that obtained with plans and/or physical models [76,77,78,79]. Videogame engines such as Unity or Unreal, or interactive systems such as Lumion or TwinMotion, and plugins such as Enscape, allow the understanding of complex models without the need for complex sensors such as HMD [85,86]. We summarize the main positive characteristics of these systems in the academic field as follows:The ability to understand the relationships between people and the architectural environment that surrounds them;The ability to facilitate the transmission of ideas and decision-making to non-expert users;The ability to apply formal, functional and technical basic principles to the design and uses of complex buildings and urban environments;The ability to design and adapt complex spaces based on the needs or profiles of end-users.

However, it is essential to note that the mere fact of using technology does not mean motivating, interesting or satisfying students or end-users, particularly in noncontact methods [87,88,89]. Using active methodologies such as gamification has provided new strategies for teachers to improve the degree of motivation, capacitation, and participation required for students without them being slaves to technology [90,91,92]. Identifying challenges and goals that may be linked to the subjects’ content and objectives [93], the student motivation is stimulated in the follow-up generating an improvement in student learning [94,95,96,97]. Combining an active methodology (gamification) and new visual technologies (such as VR sensor-based systems), we could provide a comprehensive training environment. In these environments, the information is presented from a 360° point of view, which creates an incredible feeling of presence (a stronger sensation of being inside the virtual environment than being in the real world) and a body–mind experience. Additionally, a VR environment makes it possible to avoid trainees being exposed to risky situations, the “feeling” of the impact of the wrong action, and the adaptation of smart and flexible learning environments adapted to different users and needs.

On the other hand, the inherent complexity of programming sensors and VR systems applied to educational processes leads to an inevitable abandonment of such systems. From the concept of Technological Obsolescence [98], we find studies that identify the Obsolescence of Virtual Reality Learning Environments [99]. This issue is negatively affecting the students’ opinions regarding motivation and interactivity. Thus, our study identifies this problem and allows us to see how those systems will enable the connection between applications without the need for specific programming (Enscape, Lumion, TwinMotion). These will be the ones that will end up being imposed in areas of study that do not require such specific programming for complex 3D visualization and understanding of space.

In architectural and urban proposals, visualizing the possible alternatives and obtaining the user’s opinion, we find democratic and sustainable concepts [100,101,102]. Over the years, the classic models have been replaced by computer graphics and three-dimensional CAD and GIS models (standard systems). The complex approaches to project visualization have focused on cities’ technological aspects and urban planning [103]. In the architectural field [93], they have served as mere instruments of development rather than decision-making instruments [104,105].

However, interactive visual systems such as VR have evolved. They have positioned themselves as specific technologies useful for conceptualizing objects, materials, textures, lighting, sounds and other 3D components, giving the spaces the realism that end-users are looking for [106,107,108,109,110]. We will improve the students’ spatial skills, or the teaching method applied by teachers, and improve the informal interactions of the end-user, which is a valuable complement to formal education. In this sense, another key concept to consider in these interactive environments is the customization according to the user’s profile and needs [111,112,113]. The challenges proposed by users in the gamified conceptualization of the space can help students in their education and support the success of civic participation, leading to sustainable civic engagement through collective reflection [114].

However, getting closer to the citizen is not an easy task due to an inherent lack of motivation for such proposals (usually based on the projects’ printed and graphic data). Gamification provides a bridge for this lack of motivation since it generates the necessary interest required to bring its users closer to others, which has been previously verified. In this way, the strategic choices that reshape the town can reach consensus much faster and more efficiently than previously. Students’ ability to view their designs and ideas as an endpoint at the intellectual and social levels enriches them explicitly. It helps train them in a more specific and successful way for their future jobs [115].

### 2.3. EduGAME4City 3.0. User Experience Contextualization

This project is being developed and implemented (2016–2021) in collaboration with two main entities, namely, the Barcelona School of Architecture (Catalonia Polytechnic University, ETSAB-UPC) and the School of Architecture of La Salle—Ramon Llull University (ETSALS-URL), with further collaboration with the Arts and Multimedia Engineering Departments of La Salle-URL. The main activity is limited to the university environment applying VR strategies to design 3D indoor and outdoor spaces. Nevertheless, virtual proposals are developed and tested, considering the local stakeholders’ needs and the institutions supporting the project (see Figure 1).

The main aim of the current project is to show how the implementation of “gamified” virtual strategies in architecture and urban design can increase students’ spatial comprehension and citizens’ interest in the collaborative/gamified/interactive design of spaces. To address our objectives, we have identified four primary areas of action within the project (Figure 1), namely, research, open innovation and entrepreneurship; improved education and awareness; and citizen participation [116]. Working within these four primary areas, we further identified six main actions that the project addresses, namely, education, architecture, citizenship, sustainability, territory and technology, at whose intersections we find the following five main components to develop:Art, science and technology: As we explained previously, we are developing interactive and gamified representations of urban spaces for specific proposals related to the final entities’ needs using new interactive platforms. In this sense, the most critical issue is to improve the usability and accessibility of both the proposals and VR systems with participation in fairs, exhibitions and workshops by students, teachers, professionals and citizens.Collaborative work: It is necessary to transfer data and results between neighborhood entities and administrations, thereby increasing dissemination activities at all levels, especially in the academic field, using teaching activities for the co-creation and participation of virtual spaces by students, professionals and citizens.Educational and professional actualization: The COVID-19 pandemic has changed the academic workflow such that identifying new methodologies and technologies for developing and validating architectural and urban projects has become fundamental. In this sense, we need to develop new training and content for student and teacher capacitation while also considering any data about the user profiles and the project’s final impact on the end-users.Urban planning and local management: The new interactive proposals must be developed, considering the regulations and municipal policies of tactical urbanism. All interventions in buildings, squares or neighborhoods, regarding the new spaces within the city, need understanding and the correct size design, which are aspects that smart eco-social-city technologies can address.Sustainability and social democracy: By understanding neighborhoods’ needs, we can develop projects based on co-responsibility in the design, implementation and changes of urban spaces. These actions can be applied to high-level and low-level projects, such as the design of urban furniture or the prioritization of sustainable needs.

Related to this last direction, we have identified the main Sustainable Development Goals (SDGs) that the five main components address, taking into account their definition [117], as seen in Figure 2.

To address the aims of the project, we have developed a VR interactive and gamified system that has been improved, and we have developed tactical projects concerning four main scenarios: “Superblock of Germanetes” at Barcelona; Baró Square at Santa Coloma de Gramanet; Catalunya Square at Sant Boi de Llobregat; and Pati de les Dones inside of the CCCB at Barcelona [118,119,120,121].

For each location, considering previous studies, a conceptual design, and the end-users’ needs, we have developed usability, motivation and satisfaction questionnaires to assess how the VR environment and the proposals are developed to address the main problems and needs [103]. The main results obtained for the continuous development process and the recollected user data were as follows [100]:The use of interactive digital systems in the educational process of urban design improves urban space re-evaluation.In formal and informal education related to collaborative urban design, our method improves public motivation, implications, and satisfaction in urban decision-making processes.Participants were receptive to and aware of adapting this new paradigm using advanced visualization methods.The results reflect the usefulness of the method in the academic field of architecture and define a new space-participation model guided at the local scale by single citizens and by the local community.It was demonstrated that teaching methodologies could be successfully approached using methods that adjust to student profiles and, just as important, processes that adapt to what is used in the professional field.

In the project development process, we have observed that by offering architecture students, end-users, and professionals new ways to participate in the design process, they can better visualize and understand physical projects. This positive effect allows them to develop both the dimensional and ergonomic relationships between elements as they see their designs come to life in real-time. Additionally, we have detected that professionals valued the system higher than did students. They had more knowledge of the field, a better understanding of what to do and were connected only with using the VR system. Meanwhile, the students were learning at the same time both the system aspects (what and how) rather than combining both aspects with their previous knowledge. Professionals placed more value on the systems that help them transmit problems, solutions, and ideas to both the nonspecialized and specialized public. On the other hand, students appreciated the fact that this system enables them to understand the relationships between buildings and the spaces between them. These values are associated with those who are more used to working in their careers.

In conclusion, we have identified two main gaps to address, namely, improving the usability of VR systems (RQ1) and studying why professionals score better on the system than students considering that after the implementation of the learning practices, the results in the classroom regarding motivation, satisfaction, usability are high (RQ2 and RQ3). We will address both gaps in the following sections.

## 3. Methodology

### 3.1. Optimization of Virtual Systems for a Comfortable User Experience

VR systems are considered one of the most effective means of transmitting complex information and knowledge since the elements of simulation and gamification, together with the hardware devices, can achieve full user participation. However, the penetration of VR experiences in the market only stands out in the entertainment sector and is only residual in other sectors. In this sense, part of the current challenge is developing VR systems that achieve high-quality experiences to inform, educate, and allow decisions to be made based on high-quality standard interactions in the received experiences.

Based on the previous studies developed within the GAME4City 3.0 global project, now focused on the EduGame4City sub-project and a preliminary version of a 3D space for risk prevention signal training, we have recreated a new space in Unreal Engine 4. Using the same programmed system and fixing any previously identified issues, we solved problems related to the interconnection between devices and the severe problem of sickness that was detected in the laboratory test. The original Unity project’s appearance, a maze of simple halls containing some moving spheres that the user could collect, and a flame thrower trap (see Figure 3) were also improved to provide more realism.

The aims of this phase addressed to resolve RQ1 identified in the results of the previous phases of the project were:To recreate the level into an office (with textures, assets, and lighting);To recreate the user’s path to the goal;To prepare several types of traps for the user to fall to;To give the user new abilities (being able to sprint and to call a shield that protects him/her from surprises);To prepare warning signs with different properties (such as being visible only when the player chooses to be, or having intermittent LEDs);To program collectable objects into time boons that would give the player extra time to finish the course.

As we have previously commented, to develop a 3D space, we have two main solutions using the currently more popular 3D game development engines accessible to the public: Unity and Unreal Engine 4 [122,123]. Both are free for personal use [124,125]; they are versatile, widely used in professional studios and have many free tutorials and assets available on the internet. Unreal Engine 4 (UE4) has two significant advantages over Unity, which we sought for this project: greater graphics (UE4 is widely used for architectural visualization) and a more straightforward coding method. In Unity, the programmer codes in C# scripts through Visual Studio; however, UE4, while allowing C++ coding, also gives the possibility of coding through blueprints. In UE4, a blueprint is a visual representation of the code. We can set variables, spawn components, and add inputs without the need for coding. Furthermore, unlike Unity, UE4 allows the hierarchy of assets that we need for our traps. Additionally, the navigation through the platform and the usage and placement of assets in UE4 are more comfortable and more straightforward for newcomers than those found in Unity.

However, the new problems identified in the process (and new data for RQ1) were as follows:
Exporting hierarchy: As we stated previously, a vital functionality that UE4 presents is its inheritance. We have developed an object class and then we have generated a child, i.e., an inheritance, of that class. This process’ main problem has occurred when exporting the children of a C++ class object into another project. It turns out that the inheritance of the C++ classes is more in-depth than that obtained with blueprint classes and thus is made to work specifically on the project they were created on. The solution has been to order to the engine that, if at any moment, a C++ class asks for a reference to the original project, it must be redirected to a new one (see Figure 4).User inputs: Due to the current societal conditions, we may want to avoid using intrusive inputs such as the VR headset for more hygienic and less intrusive means of control. Therefore, we have prepared the game to be played with either a VR headset, a keyboard, a keyboard and a mouse, or a joystick, both in the first- and third-person point of view (see Figure 5).Character movement: We can find three main types of character movement in this kind of game, i.e., how the character moves through the map due to input from the user:
○First, the character moves by teleporting him/herself little by little. This method removes the sickness that the movement can produce for the user and reduces some work related to preparing a smooth action; however, it adds some discomfort to the user because he/she needs to relocate him/herself after every jump.○Second, the character moves little by little at an accelerated rate until reaching a speed cap; when he/she stops pressing the forward button, the character decelerates until it stops. We found this method to be much less responsive, as it adds a lag between the user’s input and the user’s expected reaction.○Third, lineal character movement is the simplest method that reduces VR sickness the most between the three options. The user pushes the forward button, and the character immediately moves forward.
Pitfall trap: One of the traps present on the level is a hole on the ground surrounded by warning paint. We have found that the best way to prepare the hole consistently is to prepare the whole scene using BoxGeometry and then continue using BoxGeometry but with the brush typesetting set on subtractive. In this way, we can set a hole wherever we want. For the warning paint, we added another narrower cube with the warning paint material applied to it. To finish it off, we grouped the geometry to be copy-pasted as needed (see Figure 6).Particle effects: Several of the traps and boons of the level have particle effects (the flamethrower has a default flame, the wet floor has default mist), and we had at hand the default particle effects from Unreal; however, we wanted a special particle effect for the time boon (Figure 7). Therefore, we used Cascade, the particle effect creation software from UE4, to create an explosion of green “+10” displays that always face the player, which indicates to the player the quantity of time that he/she just gained.Seven segments: We needed an un-intrusive way of showing the player how long he/she has to finish the maze, so we decided to set seven segment displays in different locations around the level. Every second controls the time and sets the unneeded segments’ material to black and the needed segments to neon red (see Figure 8 and Figure 9).

After these problems were taken care of, the main usability issues identified previously for the engine were resolved. In this sense, it must be pointed out that Unreal Engine 4 provides quite a few functionalities and assets that the users may need to have already implemented, some of which we did use for the project. For example, the First-Person Character pawn is already implemented in the controllers and played with any of the market’s main VR controllers. The starter content provided us with several different particle systems, and the BoxGeometry brush settings were excellent for preparing the scenery. In conclusion, Unreal Engine 4 provided us with the tools we needed to achieve all our goals for project development and the improvement of previous usability issues that have been identified in studies and with enough documentation and forums to resolve any complications we encountered. The preparation of the map was straightforward with their tools. We could easily create new particle systems from scratch and configure the game for it to cause as little VR sickness as possible and organize the project so that anybody can modify the project and even create a new map.

### 3.2. Mixed Method Assessment

After the engine’s modifications described in Section 3.1 and solving the main problems found in the new programming process (RQ1), we conducted a mixed approach assessment [126,127]. Users were invited to play with the programmed VR system in both indoor and outdoor scenarios. This process’ objective was to familiarize the users with the environment and its possibilities, subsequently gather their feedback and identify the existing differences depending on each user’s profile. These aspects are linked to our RQ2 and RQ3. An anonymous quantitative survey and a qualitative interview for students, teachers (also professionals), and end-users were carried out.

Focusing on the quantitative instrument, we designed a questionnaire based on a Likert scale and questions that have been previously validated [120] using the Delphi method [128,129]. With the method mentioned above, we have checked which aspects of the chosen ones or questions asked are clear indicators of what they are intended to measure. The result presented is a reduction of a broader list of questions worked on in previous phases. The questionnaire was subjected to the assessment of researchers and experts (five experts in architecture education, virtual reality, educational assessment and usability) who judged and allowed us to modify the questions to ensure the instrument’s capacity to evaluate the appropriate dimensions for our study.

The participants evaluated the answers on a scale ranging from 0 to 5 and the different ICTs related to the main conceptual frameworks in architecture education and professional development. All the users were asked seven main questions about technologies and methodologies (Q1–Q7) and the final three personal perception questions (Q8–Q10). Q1 to Q3 address general perceptions about VR systems and sensors’ capacity to design and validate the structure, composition, details, and global space. Q4 and Q5 question end-user involvement in the processes of space creation and decision-making. Q6 and Q7 address personal perceptions about the user’s motivation in the use of VR systems for creating and visualizing complex spaces. The three last questions were about personal motivations and perceptions of usefulness and further training in the systems proposed. The 10 questions were:Q1—Gamified 3D systems help the user design the urban/architectural space.Q2—Three-dimensional visualization systems help the user to improve the aesthetics and composition of the urban/architectural space.Q3—Three-dimensional visualization systems allow a better understanding of complex projects.Q4—It is useful and necessary that the end-user can propose and interact with the architecture and urban spaces’ design.Q5—The proposals generated by the end-user must be considered in the execution.Q6—I am motivated to use gamified systems to design complex 3D spaces.Q7—I am motivated to use gamified systems to visualize complex 3D spaces.Q8—Previous experiences using different technologies and devices.Q9—Level of motivation to follow new training about the same systems.Q10—Level of perception of the need to master these technologies for their current and future work.

In usability and user experience research, qualitative approaches are widely used to complement quantitative methods or relatively small samples. In these types of studies, the Socratic model of postmodern psychology is also applicable and valuable because it targets highly accurate UX-related data and addresses complex information about the product, experience or technology studied [130,131]. Unlike the empirical hypothetical-deductive model, this psychological model defends the user’s subjective treatment, and it is the basis for BLA system implementation [132].

The BLA method can be defined as an exploration technique based on the psychological framework and aims to identify the critical factors of any user experience. The main goal of the BLA is to identify personal opinions about the characteristics of the element evaluated, including users’ frustration, confidence or gratitude (along with many others) [22]. The BLA technique works to identify positive and negative elements to recognize the system’s advantages and limitations. The aim of a laddering interview is to uncover how the qualities of the system, the implications of use, and personal values are connected in the mind of an individual.

## 4. Results

### 4.1. Quantitative Users’ Perception

This approach aimed to know: which methods and tools individuals are currently using, their limitations and perceived potential, their motivation to include or develop their projects using different VR engines and sensors, their reason to follow the related training, and other issues considering the VR applications based on the age, gender or profession of the sample. The sample of this quantitative approach consisted of 133 users, with distributions by age and gender shown in Table 1 and their main activities shown in Table 2.

To compare the results of the different subgroups of our sample (Table 1 and Table 2), we conducted different comparisons of the averages obtained from the initial seven questions. Considering the differences between samples and, in some cases, smaller samples for comparison, we conducted an independent t-test, which has the highest power in such cases [133]. We considered a *p* = 0.05 threshold for statistical significance of the results (two-tailed), which considers that there is a high probability of a significant difference between groups. The results are shown in Table 3 according to age and gender and in Table 4 according to the users’ main activity (related or not to architecture and urbanism).

No significant variations in gender or age were initially found (Table 3). The groupings displayed high values (all above 3.7/5) considering the Likert-type scale ranging from 1 to 5. In contrast, when we compare the findings for the users’ practices, some critical variations appear. There is a significantly higher valuation in the answers given by the group of users with activities not related to architecture. This difference appears again when we compare gender among the professionals in this sector (Table 4). If we break down the results graphically for the seven questions indicated according to the aspects that have shown significant differences, we obtain the results displayed in Figure 10.

The main differences between expert and non-expert users in the architecture field are represented, as seen and predicted, in Q4 and Q5 questions. The role of the end-user is questioned. It is observed how end-users seek the chance to impact their ideas in the final space design through contact with VR systems (with a result of 4.44/5, followed by 4.26 and 4.17). This is an idea that architects also reflect as positive but is significantly valued as less critical for realizing projects.

Equally significant is the result of the responses of users related to architecture according to gender. Females valued all the questions higher (except Q2) than did males in the same sector, with the differences for Q3, Q4, Q5, and Q7 being especially relevant, with the Q5 answers being the lowest for males; the female scores often even exceeded the positive ratings from non-expert users.

Next, we analyze the results of the answers to the questions that explicitly asked about previous experience (Q8), motivation in training (Q9), and the need for mastery for habitual activity (Q10) from the following list of technologies and systems: Virtual reality (VR); Augmented reality (AR); Interactive systems such as Lumion, TwinMotion or EndScape (IS); Video games (VR); Serious games (SG); Computer-aided design systems (CAD); Building information modeling (BIM); Photo editors and image composition (PE); Geographic information systems (GIS).

In Figure 11, we can observe how the dominant technologies are still the most traditional, i.e., those that base project creation on CAD systems and their simulation on the photomontage of designs, photographs and renderings (PE). The low use/knowledge of BIM systems by architecture-related users is shocking (overall 2.86/5, with a significant gap between males (3.21) and females (2.50)), especially given that these BIM systems are the ones that are reflected as being of the most significant current importance in the profession (see Figure 12), with an average score of 4.54/5 among architects. In this same sense, the GIS systems also stand out, with very low levels of preparation and habitual use but with a great need for professional use/implementation (obtaining a score of 3.88/5 and ranking behind CAD, PE, and BIM systems).

It is also observed that virtual, digital, immersive and/or gamified technology-based applications globally do not have a very high user experience with either type of user profile. At the same time, there is a statistically significant difference between architecture-related users (Av: 2.61, SD: 0.03) and non-expert users (*p* = 0.0020, Av: 3.21, SD: 0.05). This significant difference disappears when we evaluate the importance of these systems regarding the current tasks (see Figure 12). As seen, the importance of systems based on interaction (AR/VR) rises globally to values in the range from 3.5 to 4 according to the current need, without the gamified approaches undergoing major changes.

The importance reflected in AR/VR systems and the general importance of all those interactive systems are very relevant, since globally, IS obtains a score of 3.74/5 according to the abovementioned systems and is thus as common as GIS (3.22) and very close to both traditional methods (CAD: 4.08 and PE: 4.07) and the current BIM (3.95). This increase in importance is not transferred to gamified systems, which are perceived as being of less importance in professional tasks, which is a fact that collides with the trends that we have previously referenced. This shows how gamified strategies are good systems for approximating complex projects both at a training level and in interactions with end-users.

Finally, in this quantitative exploratory phase, we analyzed the users’ motivations in their possible future formation of the identified systems (Figure 13). Globally, there were no significant differences between the motivations for users’ training (*p* = 0.2221). However, after analyzing the systems separately, we can confirm that for interactive and gamified systems, the behavior among expert users within the architectural field, students, and non-expert users is homogeneous (*p* = 0.5309, with an Av: 3.63, SD: 1.18 for architecture users, and Av: 3.83, SD: 1.25 for non-architecture users), while in terms of systems related to professional use, a significant difference between the profiles is demonstrated (*p* = 0.0124, with an Av: 4.20, SD: 1.08 for architecture users and Av: 3.29, SD: 1.54 for non-architecture users).

In addition, the previous but complementary findings obtained for a study based on professors and specialists in the architectural field must be taken into account [134].

### 4.2. Qualitative Bipolar Laddering Assessment (BLA)

There are three phases of the BLA method: recognition, assessment, and suggestions. In the first phase, regarding the visualization of architectural and urban spaces, the user determines their VR interaction’s positive and negative aspects with HMD sensors. In a second phase, using a score between 0 (very negative element) and 10 (very positive element), the user labels his/her identified positive and negative ideas. Finally, in the third phase, the user proposes or recommends changes to each concept (positive or negative) to strengthen it in the future.

From the results obtained, we polarized the answers into positive elements (common positive elements, i.e., elements cited by more than one user (CPx), and particular positive elements, i.e., elements mentioned by only one user (PPx)) and negative elements (represented by CNx for common negative elements and by PNx for particular negative citations). In Table 5, we can observe the resumption of the positive aspects identified. In Table 6, we can observe the negative aspects, considering that we conducted 16 interviews with the following distribution characteristics: eight females (Age Av: 38, 38) and eight males (Age Av: 44, 38). Nine users were related to architectural activities, and only seven users had previous experience with VR systems as users.

As seen from Table 5 and Table 6, there is a high consensus on the positive and negative aspects identified, with most aspects being cited by two or more people (common aspects). In a positive sense, with an MI of 75%, the spatial comprehension that systems based on HMD sensors provide (CP1) stands out. Significantly lower, but with rates above 30% and an average valuation of approximately 9/10, the capacities to add extended data to the elements of the virtual environment (CP3), the interaction (CP2), and the possibility of detecting errors and problems in spaces (CP5) are significant positive issues that are identified. At the opposite extreme, for identified negative issues, no main significant problem is identified as standing out. Still, there are three main problems with an MI of at least 25%, namely, lack of habit in the use of these systems (CN6), accessibility and costs that they entail (CN4) and the lack or loss of physical interaction with objects (CN8). This concept is specially referred to by users with an architectural background.

The table of solutions and/or enhancements found by the users is shown in Table 7 and the aspects identified. The most cited aspects (above the 50% sample), as seen, are in line with our assumptions. There are two clearly identified aspects: the need to produce further examples to specifically understand the possibilities of VR systems applied to any area (CS3) and add more time for both academic and professional (CS6) practice and tests. These results reinforce the need to continue to develop the knowledge that integrates these conditions so that they are more comparable with reality and their needs (CS3).

## 5. Discussion and Conclusions

Architectural and urban learning and training are currently focused on using PBL approaches for both technical and project subjects. This methodology is most widely found in face-to-face environments, with conventional lessons and discussions available during workshop time. Despite the current hybrid sense emerging from the consequences of the COVID-19 pandemic, such learning generally requires a different level of usage and help from technology. The technical subjects need to modify their workflow using interactive and virtual systems for deliveries and corrections. On the other hand, the project subjects without the possibility of managing printed plans or physical mock-ups need new methods to review, present and discuss the new proposals, which is something that VR systems can support.

By analyzing the results obtained from the improvement phase of the system and its assessment and considering our research questions, we can summarize:

RQ1: What are the main problems to be solved in VR sensor programming using the main videogames such as Unity or Unreal?

In this sense, we have identified two main problems: sensor programming and usability. With the exporting hierarchy technical issues the solution has been to order to the engine that, if at any moment, a C++ class asks for a reference to the original project, it must be redirected to a new one. Additionally, we have grouped elements for pitfall traps, and we have used the particle effect creation for special effects that Unreal offers. For usability issues, we have set up the game to be played with either a VR headset, a keyboard, a keyboard and a mouse, or a joystick, both in the first- and third-person point of view. For improving the sickness, we have programmed a slow movement associated with a specific command for the character movement. Finally, we have added seven segment displays in different locations for helping the user’s interaction.

On the other hand, it is necessary to improve the usability of the system. Our next steps in this direction are as follows:Catalogue: The first version of the project had a catalogue with nine assets available. The labels had to be manually and individually prepared, and the spawned assets linked to their icon. In the future, we will prepare a catalogue that displays a preview of all available assets and shows them in 3D.Import assets: Once the platform is ready, we want the user to be able to load new assets for later placement within the environment. The user will only have to place the 3D objects into a folder, and the program will automatically load them all. If possible, the items will be divided into folders within the catalogue.Lights: The user will be able to place assets and lights and regulate their intensity and color.Light switches: The user will be able to link lights to switches and later use them.Real sun position: In the last two versions, the user could rotate the sun position at will. This time, the user will be able to set a place and a time, and the sun will be set at the correct position.HUD: Since we will add many functionalities, it will be wise to prepare dynamic HUDs to show the assets’ properties for the user to choose from.Save and load: The user will be capable of saving a scene to load it later.Camera: The user will be able to export renders of a scene, as if he/she had a camera on him/her.Blueprint generator: In case the user wants to check the scene from a top-down perspective, he/she will be able to generate a render of the environment as if it was a construction blueprint.Wall generator: In case the user wants to add some new edifications and new buildings but does not have enough assets, he/she will be able to generate walls through the scene and adjust their dimensions as he/she sees fit.Material modifier: Following the previous line, the user will be able to change the assets’ textures whenever he/she wants.Multiplayer: The platform may be useful for an architect to show the environment to a person. However, we may also want both individuals to move at the same time through the scene. That is why we will develop a multiplayer mode.

Considering the results obtained from our mixed assessment, which are related to our research questions 2 and 3, we can conclude:

RQ2: Are there differences in use, perception, satisfaction and/or expectations of serious games implemented in VR systems in the function of the user’s profile?

The outcome of the survey shows a higher degree of awareness and interest in using VR systems by non-expert users and, in the case of architectural professions, by females. This feature is interesting to the point that it establishes possible links between the following statements:There is resistance to the use of advanced visualization systems in the architectural teaching field because most teachers (males) are more reluctant to use non-traditional systems such as printed plans, models, or panels;This resistance may be due to the previous teachers’ education, which is based on traditional methods, and their lack of training regarding the new method;This outcome is reflected at the educational level, where, due to the majority of male teachers [135,136], the use of VR or interactive systems is not prioritized in educational practice.

In summary, there is a gender difference that affects the use of VR to understand space, which is a difference that has been previously identified in other studies at the general level [137,138]. In this sense, all the STEM/STEAM initiatives of the last decades advance an improvement of the results as the more technical vocations among females gain more space and the number of teachers and professionals becomes equal in terms of gender and even becomes renewed at the age level [139,140,141,142]. The differential behavior by gender and age of such professionals is evident, with a tendency to be more motivated to be trained and to include specific systems in the academic field by young female professionals. As we have shown in our study, the current need for the use and domain (VR and interactive systems) is very strong at a professional level. Still, older and male users are more unwilling to train and include these techniques in the academic field.

Analyzing the qualitative sample, the highest results, both positive and negative within the expected answers, reflect the great potential that all users perceive. However, a lack of VR system knowledge is present, especially in terms of the difficulty that entails its implementation both didactically and professionally. In this sense, the identified aspects show and reaffirm the idea of resistance to implantation based on a series of prejudices. Although there are no direct bridges that allow the linking of CAD and BIM systems with virtual environments, plugins and modules enable them to be fitted into video game engines (such as Unity or Unreal) without excessive problems.

RQ3: Why are the immersive and/or gamified systems poorly used in architectural education when their potential has been widely demonstrated?

The following conflict is reflected in the two participatory processes: professors at the professional level are using classical methods (CAD and photographic composition). However, they reflect a growing motivation regarding advanced systems (especially the younger professionals and women); this interest has not been reflected in its use within subjects. The perception that we are dealing with expensive systems that are difficult to use and poorly explained causes an unconscious response towards their use, strengthening the lack of use. The assessed solutions specifically showed their potential to aid in the interpretation of the environment and the making of decisions based on perceived needs, also in the educational workflow [143,144]. Applying additional data to three-dimensional and immersive models is an intrinsic feature of architectural practice; however, it is subcontracted at a professional level and derived to technical subjects at an instructional level, without implementation at the project level.

Due to this lack of use, students do not perceive the usefulness of such advanced tools within their training or gain a sense of them as complex training. While AR, VR, 360° interactive viewings, gamified proposals, etc., are present in the main professional recommendations, they have only a minimum level of technical subjects’ implementation. There is a high level of interest (in all levels and types of users) in learning and applying these methods [145], but there is still a high volume of older teachers. They are professional experts and teachers in the fields related to the project. Still, they have not been updated in the new interactive, parametric or visually more advanced systems that help streamline the architectural project/urban co95mplex workflow.

To improve and include new learning approaches using new visual systems, such as VR interactive methods or/and gamified proposals based on user interaction in VR environments, divisions between subjects must partially disappear. The quality of deliveries developed using VR sensors and frameworks generates a great space’s comprehension, being a powerful tool to simulate complex, real situations and environments, offering researchers and teachers unprecedented opportunities to investigate user behavior in closely controlled designs in controlled laboratory conditions, both in end-users and students and professionals [146].

Taking the PBL methodology as a common approach for technical and project subjects, there is a need to merge the same projects and develop new digital deliveries. As has been established, all implementations need further explanation and time to extend these programs to further tasks and presentations. Each user’s needs can differ; in these situations, technicians’ assistance in adjusting each delivery to the final user is required. In this context, this is one field where VR has been much applied. Previous works [27] showed that VR could offer significant educational advantages. It can solve time-travel problems, and, for example, students can experience different historical periods, non-constructed spaces, special conditions, and other risks and situations. It can address and test the overall validity of proposed urban plans and architectural designs, generate alternatives and conceptualize learning, instruction and the design process itself [147].

It has been shown that teaching methodologies could be successfully approached using methods that adjust to a student’s profile and just critical methodologies that adapt to the items used in the professional field. Nevertheless, an aspect that needs to be evaluated in the future is that the participants gave a lower value to the statement that they would use the program again in the future and explore the gender and age gap. Although today’s students fit the profile of individuals who are comfortable using technology to interact and represent ideas, there is still a difference between the ability of ICTs in the classroom and their practical application in the workforce. It is essential to improve how these methods are introduced and clarified in educational institutions to close the divide between academic fields. The latter seems to be more prepared to incorporate all kinds of technologies, interaction, gamification and different strategies. In this sense, it seems clear that our researcher needs to re-force the training in all levels, during the studies and the continuous training in professional life [148].

It is also crucial for the continual development of the programmed processes to incorporate new directives, setups, and adaptations for material awareness, facility management, structure measurement, risk management and preparation, urbanism, and other fields. Interactive visual systems improve the efficiency of workflows and the sustainability and production costs of the architectural project, especially in highly virtualized environments with low levels of face-to-face interactions, as demonstrated in the pandemic situation, where training elements such as models and printed plans have given way, for the first time in a general manner, to digital deliveries and other types of interactive deliveries.

## Figures and Tables

**Figure 1 sensors-21-03102-f001:**
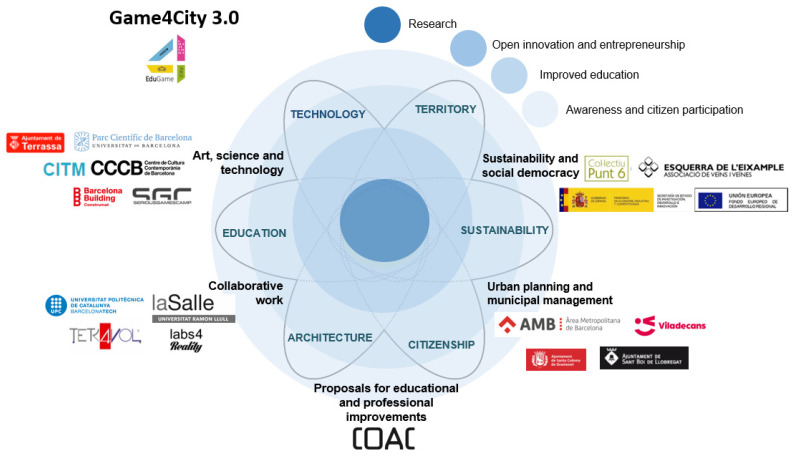
Global Game4City project scopes, lines of action and agents. The project has two sub-projects: ArchGame4City developed by UPC, and EduGame4City developed by URL.

**Figure 2 sensors-21-03102-f002:**
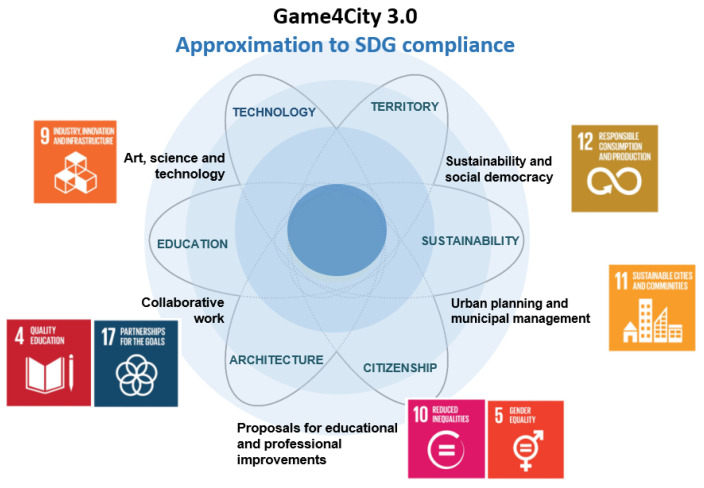
Game4City project scopes, lines of action and their relationship with SDGs.

**Figure 3 sensors-21-03102-f003:**
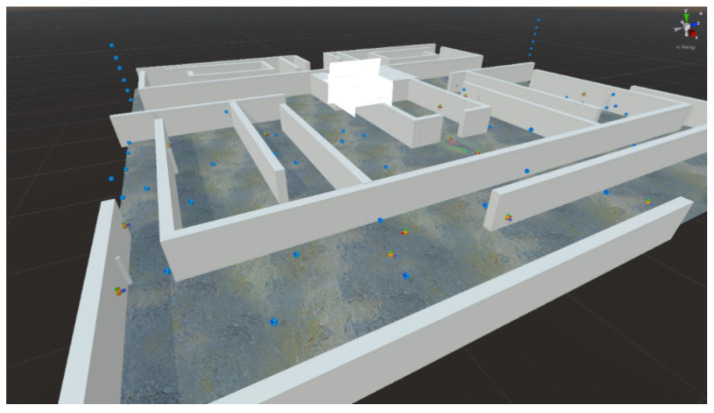
Initial configuration programmed in Unity of the space created for risk prevention and signal training.

**Figure 4 sensors-21-03102-f004:**
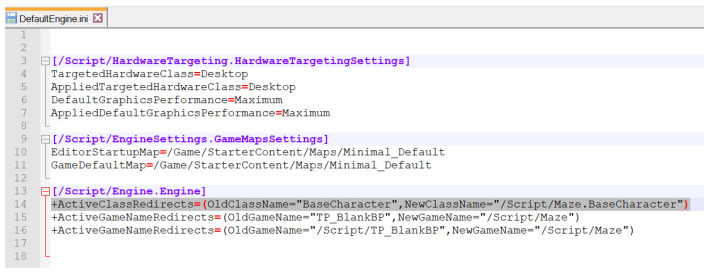
Solution for exporting the hierarchy problem.

**Figure 5 sensors-21-03102-f005:**
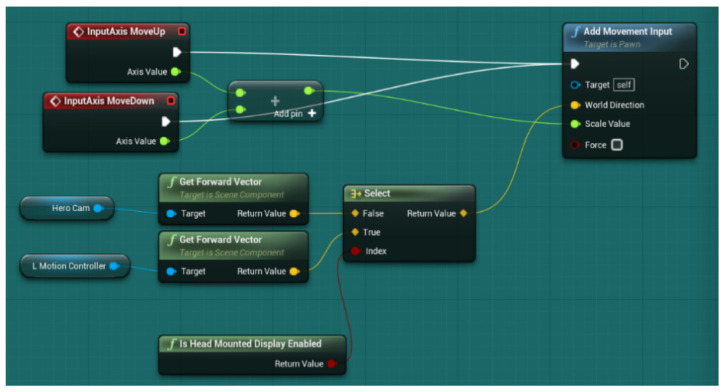
Solution for different user input devices.

**Figure 6 sensors-21-03102-f006:**
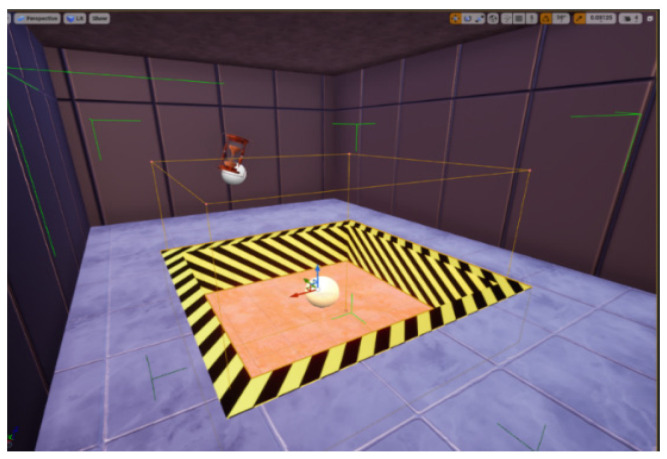
Example of the pitfall trap solution.

**Figure 7 sensors-21-03102-f007:**
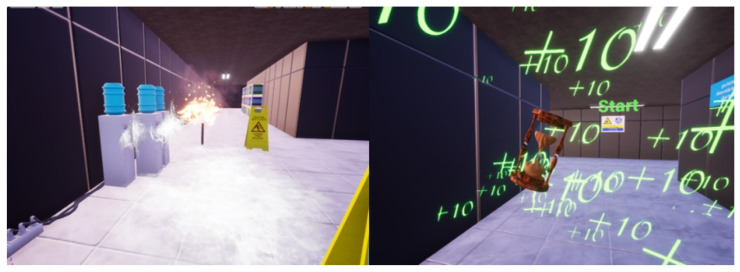
Solution for particle effects using Cascade.

**Figure 8 sensors-21-03102-f008:**
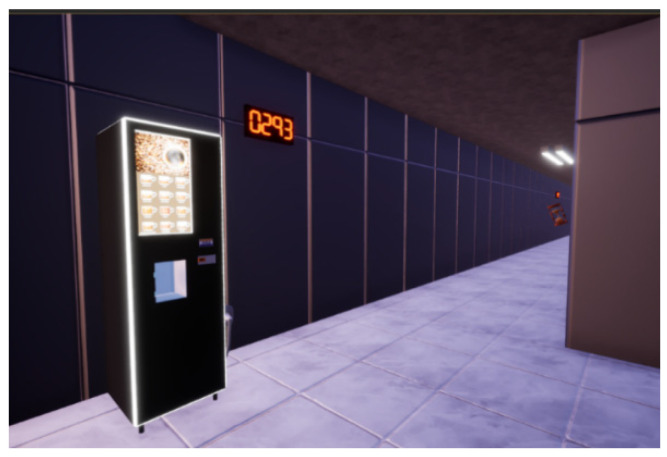
Example of seven-segments solution.

**Figure 9 sensors-21-03102-f009:**
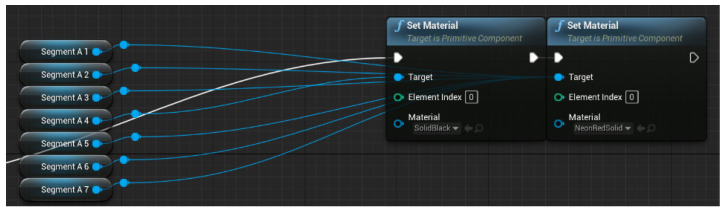
Seven-segments displays set up.

**Figure 10 sensors-21-03102-f010:**
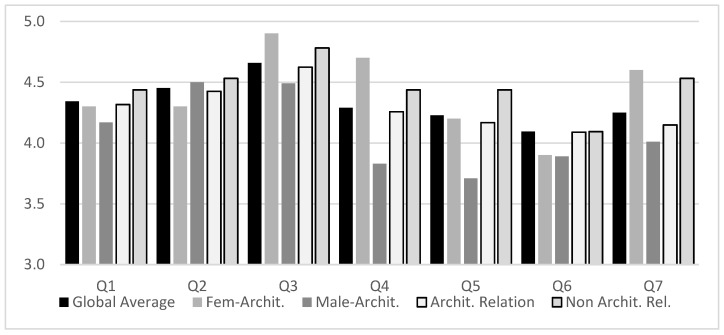
Q1–Q7 Results: Global average by gender related to architectural activities and comparison between non-architectural and architectural professionals.

**Figure 11 sensors-21-03102-f011:**
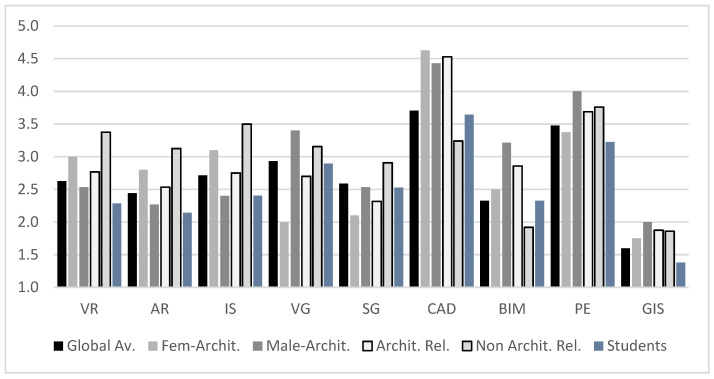
User experience with the technologies cited.

**Figure 12 sensors-21-03102-f012:**
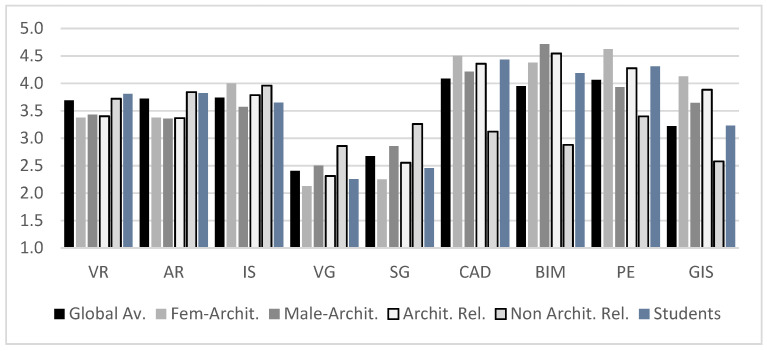
Level of importance of these technologies in current tasks.

**Figure 13 sensors-21-03102-f013:**
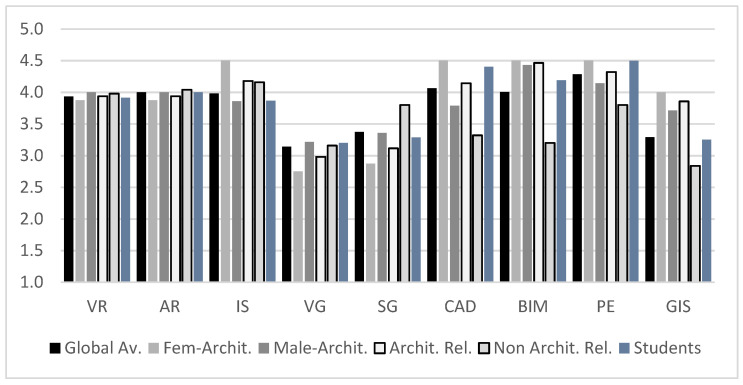
User motivation in future training.

**Table 1 sensors-21-03102-t001:** Sample data by age and gender.

Sample by Age and Gender	Females	Average (Av.) and Standard Deviation (SD)	Males	Av. and SD
18–24	49	20.02 (1.56)	29	20.20 (1.67)
25–40	11	30.18 (5.11)	13	29.76 (2.80)
41–49	8	45.25 (3.05)	11	45.81 (3.48)
More than 50	6	54.00 (6.66)	6	55.16 (5.19)
GLOBAL	74	27.01 (11.82)	59	30.64 (12.96)

**Table 2 sensors-21-03102-t002:** Sample data by main activity developed and gender.

**Sample by Activity**	**Arch.**	**Av. and SD**	**Non-Arch.**	**Av. and SD**
Archit. vs. No architectural activity	100	24.26 (8.95)	33	42.41 (11.87)
	**Females**	**Av. and SD**	**Males**	**Av. and SD**
Students	47	20.46 (2.84)	28	20.71 (3.14)
Architecture professional activity	10	32.30 (10.59)	15	37.67 (11.84)
No architectural professional activity	16	43.38 (12.08)	15	42.33 (11.82)

**Table 3 sensors-21-03102-t003:** T-test two-tailed (p) results of Q1–Q7 comparison, by age and gender.

Sample by Age and Gender	Av. Female (SD)	Av. Male (SD)	*p*
18–24	4.34 (0.02)	4.31 (0.01)	0.7318
25–40	4.35 (0.08)	4.18 (0.08)	0.3322
41–49	4.48 (0.12)	4.41 (0.08)	0.7107
More than 50	4.45 (0.13)	4.50 (0.09)	0.7967

**Table 4 sensors-21-03102-t004:** T-test two-tailed (p) results of Q1–Q7 comparison, by activity.

Sample	Av. and SD	Av. and (SD)	*p*
By activity (Arch vs. Non-Arch)	4.20 (0.03)	4.46 (0.04)	0.0317
Architecture students (fem vs. male)	4.35 (0.03)	4.36 (0.01)	0.7793
Architecture professionals (fem vs. male)	4.41 (0.11)	4.05 (0.09)	0.0422
Non-architectural users (fem vs. male)	4.41 (0.08)	4.60 (0.04)	0.1897

**Table 5 sensors-21-03102-t005:** BLA positive elements of HMD-VR systems identified.

Item Id	Description	Av. Score (Av)	Mention Index (MI)
CP1	Space comprehension	9.08	75.00%
CP2	Interaction	9.20	31.25%
CP3	Extended data (lights, materials, other)	8.33	37.50%
CP4	Student capacitation in new ICTs	9.33	18.75%
CP5	Error/Problem detecting	9.00	31.25%
CP6	Multiple uses	8.00	12.50%
CP7	Professional uses	9.25	25.00%
CP8	Better communication with final users	9.50	25.00%
CP9	Initial comprehension	9.00	12.50%
PP1	Scale 1:1 working	9.00	6.25%
PP2	Improve motivation in the users (Stud. and Citz.)	7.00	6.25%
PP3	Accessible technologies	8.00	6.25%
PP4	Game options	9.00	6.25%
PP5	Cost	9.00	6.25%
PP6	Quick System	7.00	6.25%

**Table 6 sensors-21-03102-t006:** BLA negative elements of HMD-VR systems identified.

Item Id	Description	Av. Score (Av)	Mention Index (MI)
CN1	Individual use	4.67	18.75%
CN2	Technology comfort	5.67	18.75%
CN3	Visualization and sickness	3.00	12.50%
CN4	Accessibility/Cost	4.00	25.00%
CN5	Lack of continuous use	4.00	18.75%
CN6	Lack of habit	3.33	37.50%
CN7	Non-professional support/less critical thinking	4.00	18.75%
CN8	Non-physical/real interaction	3.00	25.00%
CN9	Difficult to use	2.33	18.75%
CN10	Lack of standards	2.50	12.50%
CN11	Moment of use	2.67	18.75%
CN12	Developing time	6.00	12.50%
PN1	Lack of other senses	3.00	6.25%

**Table 7 sensors-21-03102-t007:** Common and particular solutions and improvements for positive and negative elements were identified.

Item Id	Description	Mention Index (MI)
CS1	Capacity of interaction with other users (Multi-user)	25.00%
CS2	More explanation to understand more possibilities of VR systems	56.25%
CS3	More quality/data (real materials, textures, lights, etc.)	56.25%
CS4	More interaction with the space and objects	37.50%
CS5	Improve usability of VR Glasses (new devices more accessible)	18.75%
CS6	More practices in the academic/professional fields	62.50%
CS7	No limitations in the field of project presentation/visualization	12.50%
CS8	Improve the interaction and sharing ideas with the final users	37.50%
CS9	Improve the timing of project development	25.00%
CS10	Adaptation of the VR environment based on the user profile	37.50%
CS11	Use of disposable and cheap devices	18.75%
CS12	Add/improve link with other devices (touch, hearing…)	31.25%
CS13	Improve standards between software/hardware	31.25%
CS14	Share the control between technological and project subjects	31.25%
PS1	Improve its uses in interior spaces	6.25%

## Data Availability

The data of the study can be accessible at this link: https://lasalleuniversities.sharepoint.com/:f:/r/sites/doa-ddr-gretel/Shared%20Documents/2021-Publications/Sensors-David-VR?csf=1&web=1&e=UXHCbK (accessed on 27 April 2021).

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
