# Peer review of "Mixed Assessment of Virtual Serious Games Applied in Architectural and Urban Design Education"

_sensors, 2021, doi:10.3390/s21093102_

Round 1

Reviewer 1 Report

The authors have presented an interesting work about the usefulness of mixed reality in urban design.

Some problems must be fixed before publication.

  • The introduction fails to describe the novelty and how it is different/similar regarding related papers and commercial softwares.
  • Section 2.1 shows a generic text about AR and VR and does not enhance the proposal. 
  • Sections 2.2 and 2.3 are the same. Already know information without defining the methods and the detailed proposal.
  • Section 3 is the results.. but the methodology was not even discussed nor adequately presented. Most of this section is not result, is methodology.
  • The actual results are not well discussed.

All in all, it is a good paper, may have value for publication. However, the writing and academic design were not properly done. The paper is too prolix and with a lot well know information that does need to be there. The research is not well presented, and the results are not well discussed.

The authors must keep in mind the journal's focus (sensors), so they should have a clear statement of the environment, the goals, how to collect and analyze the information. 

I do believe it is a good paper, but it must be redesigned from a more objective point of view.

Author Response

Dear reviewer, thanks for your time, review and suggestions. We have improved our manuscript following all the suggestions and ideas received. Please find attached our detailed responses.

Best 

Reviewer 2 Report

Sensors-1169101: Mixed assessment of virtual serious games applied in the architectural and urban design of learning processes.

The authors address an original research work, which is a rigorous and well-organized paper. Although this topic is not a usual one in the journal Sensors, I am grateful to have been a reviewer of this paper because I found it very interesting. Furthermore, I think it has a great potential to have a good level of citation.

I just wanted to make one recommendation to complete this research work. Recently, a previous paper highlighted the influence of technological obsolescence on virtual reality learning environments (https://doi.org/10.3390/app10030915; https://doi.org/10.1016/j.techsoc.2020.101347). In my opinion, the authors could include a mention of this influence in the manuscript.

Author Response

(The authors gave the same response as above.)

Reviewer 3 Report

Dear Authors,

I really liked the project which you described; however, your paper concerning the project research activities requires extensive editing and restructuring.

My first thought is that in this shape, the paper does not fit the journal. Therefore, it needs rethinking and further elaboration. Please consider the structure (specifically, the aim, research questions) and style (long, rambling sentences appear many times).

First of all, the aim of the article is unclear. You write that "this paper illustrates the improvement of a virtual navigation system through user studies carried out (...)". Such a description is broad. Maybe you could think of a more precise aim? Next line is about the project's aim. But do you refer to the project, or to the article? 

Secondly, the text contains a lot of weight-end sentences, for instance: "Based on the research results, weak points regarding this kind of gamified interactive system that must be rectified to improve its adoption in academia have been identified" [lines 26-27]. Long, complex sentences could be rewritten and simplified.

Also, the title is a bit confusing: "Mixed assessment of virtual serious games applied in the architectural and urban design of learning processes". Maybe it would sound better as: "Mixed assessment of virtual serious games applied in the architectural and urban design education"? [in the line 42 you introduce the issue of education].

In the line 53, instead of "The paper's specific scope (...)", maybe the aim of the article should be provided?

In the lines 56-57, do you mean that the 3D architecture elements are built to serve as a didactic aid? If yes, could you write it more explicitly?

In the line 63 you refer to objectives: "To assess the objectives of our research (...)". How many objectives are there? Where are they in the text? 

In the lines 63-64 the word "sensor" appears for the first time. The journal's scope is related to sensors https://www.mdpi.com/journal/sensors/about and this could be the keyword addressing to your investigations. Maybe you could rethink this idea and design of the paper in the way it will reflect the journal's scope. In the present design, the "sensor" seems to appear accidentally... . Both the title and the abstract (at least abstract only) could refer to "the sensor" issues.

Part 2.1. Sensors and main problems using virtual systems [line 76] is not about sensors. It is about VR. Why do you mention AR at the begin of the section (in a topic sentence), if it is not further elaborated?

Lines 83-84: could you avoid generalizations, such as "innovating certain practices"? Or refer to more specific examples? 

All chapters and sections should be rearranged. Methodological part should explain what you did and how you did it, allowing a reader to evaluate the reliability and validity of the research (the type of the research, how you collected & analysed your data, any tools or materials you used in the research, your rationale for choosing these methods). All in connection with the use of a sensor in your project. The full experimental details must be provided so that the results can be reproduced.
Also, a novelty of your approach and practical implications should be described.

References and the content do not reflect the interests in sensors sufficiently; they should be improved. 

Regarding all issues above, I strongly recommend you to take your time and rethink the whole content. The idea of using a sensor in UX research for educational purposes is really worth sharing with the readers.

Kind regards.

Author Response

(The authors gave the same response as above.)

Round 2

Reviewer 1 Report

the authors have done a good job.

Author Response

Dear Review

Attached the responses to your review... 

Best. 

Reviewer 3 Report

Dear Authors,

thank you very much for improvements. After reading a new version of the article, I found two major issues which should be resolved.

Regarding the research design, could you provide information about the sensor in the abstract and in the introduction? In the Introduction, the concept of sensor appears for the first time when you describe the novelty aspects. For a reader it would be valuable to explain what kind of a sensor/sensors you use in your research and why. Also, what types of activities are measured? Can you analyse literature in this scope? 

Line 50: "The paper's novelty lies in the identification and improvement (based on the results of previous phases of the project) of the critical aspects that affect the usability in the interaction with Virtual Reality (VR) sensors": could you specify what is going to be improved? (What do you expect to achieve?) This explanation is too enigmatic, also its further description is too general.

Line 58: "(...) another novel aspect of the paper is the mixed-method research approach used in the evaluation." Is mixed-method research a real innovation?

Line 66: Unity or Unreal are not "videogames", but a game engine (maybe it would be valuable to explain it a bit).

Line 566: how was the tool validated?

Do you have any data concerning a statistical significance of sensors' parameters, what functions were determined between variables. Literature analysis would be necessary in this scope. 

The conclusions concentrate on the system architecture, new functionalities of game engine and different user interfaces (keyboard, etc.). Could you emphasize the role of sensors in your study?

All in all, the article could explore more the issues of VR, sensors, and human reactions. 

Author Response

Dear Reviewer 

you can find attached our responses to your comments and suggestions.

Best. 
